# PROACTIVE LEARNING: SEARCH-AUGMENTED LEARNING USING PRE-TRAINED MODELS

## ABSTRACT

Large pre-trained models are increasingly important tools in machine learning. Although versatile, their knowledge is however limited and often insufficient to handle domain-specific nuances. This paper introduces a novel "webly-supervised" learning approach that uses web search to enrich a pre-trained model for visual recognition. This strategy empowers the model to access relevant, up-to-date information as required. Our method first identifies test instances that the pre-trained model is uncertain about. We then formulate a query for Google Search to retrieve images to resolve this uncertainty. These serve as noisy data to train a compact classifier, with no need for additional manual labelling.

While multiple attempts at search-augmented learning appeared in the past, this iteration of the concept benefits from recent advances in NLP and multi-modal learning. This allows demonstrating unique benefits in uncertainty quantification and domain-specific recognition (*e.g.* +15 percentage points in accuracy on the Stanford Cars and Flowers datasets). We also present extensive experiments to explore the impact of noisy retrieval and different fine-tuning strategies.

## 1 INTRODUCTION

The World Wide Web is the source of most training data used to train today's largest models. The amount and diversity of data available has been key for training the best models for vision, NLP, and multimodal tasks (*e.g.* DINO (Caron et al., 2021), GPT (Brown et al., 2020) LLama (Touvron et al., 2023)), Flamingo (Xu et al., 2021)). Large pre-trained models have shown remarkable versatility in the tasks, domains, and settings they can handle. But the models' breadth often eclipses a number of limitations. Data quality and curation remains important (Fang et al., 2022; Nguyen et al., 2022). And increasing dataset sizes only masks the models' failure to generalize out of distribution (Teney et al., 2023). Increasing the size of models and datasets also does not seem sufficient to prevent the models from memorizing biases and spurious correlations (Lee, 2019)

To address the limitations of large pre-trained models, one increasingly-popular approach is retrieval-augmented inference (Teney & Hengel, 2019; Guu et al., 2020a; Shuster et al., 2022). This concept retrieves instances at test time to help with generalisation. Existing methods rely on a supplementary (*e.g.* up-to-date or domain-specific) dataset, from which the model can retrieve information to supplement its pretrained knowledge. Identifying relevant instances is challenging, these methods incur increased computational cost at test time, and the limitations related to finite datasets remain (Wang et al., 2020; Xiong et al., 2021).

In contrast to the fixed static datasets of machine learning, humans continuously update their knowledge using new information, often acquired by querying the web through search engines. Could machines do the same? The idea of connecting learning algorithms with the web and search engines is not new but is challenging to implement effectively (Carlson et al.; Chen et al., 2013; Chen & Gupta, 2015; Shen et al., 2018; Teney & Hengel, 2019). This paper present a novel iteration of this concept using recent advances in NLP and multimodal learning (Jain et al., 2022; Girdhar et al., 2023; Dong et al., 2023).

This paper explores the synergy between pre-trained models, which capture rich concept priors, and search engines, which have evolved to retrieve diverse and representative information based on text queries. We propose a methodology in which a small set of retrieved examples from the search

Figure 1: We propose a method to enrich a pre-trained visual recognition model using web search. At test time, we determine the model's uncertainty about the given input and candidate labels. We then issue a text query to acquire additional data, which we use to update the model. The predictions of the original and updated models are compared to assess the information gained and generate the final output.

engine is used for fine-tuning the pre-trained model for improved predictions (see Figure 1. Since fine-tuning generally entails making only minimal updates to model parameters (Hu et al., 2022), we proceed to train a compact model tailored for the task using this retrieved set.

Contemporary to our efforts, Li et al. (2023) recently proposed Internet Explorer, a method for representation learning that also involves an active exploration of the web. Their approach, formulated as reinforcement learning, samples concepts from ConceptNet (Speer et al., 2017) to uncover potentially-relevant images. Our approach aims to be much more frugal and efficient in the retrieved data. We use a more *directed* search strategy, and the process is only triggered when the pre-trained model shows high uncertainty.

Our approach distinguishes itself from traditional active learning (Settles, 2010; Parvaneh et al., 2022) where the objective is to minimize the number of labels requested from annotators, given a fixed set of candidate examples. In contrast, our approach seeks new instances by formulating a textual search query. It then uses them as noisy, weakly-supervised examples to improve the model.

Technically, we identify concepts whose representation in the pretrained model (CLIP) could be enhanced, by examining the distribution of image projections onto the unit hypersphere. Interestingly, we observe that instances with low predictive entropy (*i.e.* deemed certain) remain largely unchanged after fine-tuning with retrieved images. This suggests that the pre-trained model may require adjustments in its conceptual decision boundaries to better handle nuanced instances and accurately classify the desired classes. Furthermore, we find that using class descriptions for uncertain instances, as opposed to generating instance-level queries from captions (Li et al., 2022b) results in better performance. This underscores the idea that current benchmarks primarily emphasise generic class-level features, which are complementary to the information that can be gathered from web searches. Our investigations also point at limitations of existing benchmarks for assessing the capabilities of large-scale vision models (see results with the Food dataset, Section 5.2).

In summary, the contributions of this paper are as follows.

• We propose an novel approach for enriching a pre-trained visual recognition model through test-time access to web search with no need for additional labels or manual inputs. We show this approach can easily be integrated into the current machine learning pipelines and leads to improved performance.

• We implement the method on top of CLIP and derive a measure of uncertainty based on the classification entropy to make the method efficient in the amount of retrieved data. Technically, we use the projection of image embeddings onto a hypersphere, and characterize the distribution of underlying concepts using a mixture of von Mises-Fisher distributions.

• We conduct extensive experiments to explore various implementation choices. We demonstrate significant improvements on the Stanford Cars (Krause et al., 2013) and Flowers (Nilsback & Zisserman, 2008) datasets with remarkable improvements of over 15 percentage points in accuracy.

## 2 RELATED WORK

**Using unlabelled or weakly labelled data.** One area that shares some similarities with the proposed approach is active learning Settles (2010). In this paper, instead of relying on human experts, the

proposed approach leverages the vast knowledge source available on the internet. The proposed approach also connects to weakly supervised learning, where the aim is to learn from data that is only partially labelled Zhou et al. (2018). In this case, the data collected from the search engine provides weak supervision for enhancing the model's performance. While weakly supervised learning often focuses on improving the quality of the labelling, the proposed approach aims to leverage the search engine as a source of inductive biases to improve the model's predictions.

**Cross-Modal retrieval.** Cross-modal retrieval is another area that is relevant to the proposed approach, where the goal is to retrieve relevant data from different modalities such as text, image, and video Jiang et al. (2017). While cross-modal retrieval typically focuses on retrieving data from a fixed repository, the proposed approach enables real-time, up-to-date, and relevant information to be retrieved from the internet using natural language queries.

**Language models.** The success of the proposed approach also relies on recent advances in language models, such as OpenAI's GPT-3, which enables machines to interact and process natural language text Brown et al. (2020). Jointly learning image and text in models such as OpenAI's CLIP Radford et al. (2021) or Google's ALIGN Radford et al. (2021) have shown promise for improving the performance of visual models. In addition, self-supervised learning and generative modelling have shown promise for learning from noisy labels Chen et al. (2020). By utilizing these technologies, the proposed approach takes a significant leap forward that otherwise would not be possible.

**Retrieval-augmented.** Retrieval-augmented learning offers an exciting avenue for enhancing the capabilities of models in an era where access to vast and dynamic information is paramount. For instance, Guu et al. (2020b), Guu et al. (2020a), Shuster et al. (2022) learn to leverage large additional data sources for predictions. However, that requires using a common embedding space and implementation of indexing mechanisms.

**Webly-supervised learning.** NELL Carlson et al. (2010) and NEIL Chen et al. (2013) are amongst the first to explore acquiring new concepts and relationships that are periodically being refined with human supervision from the web. Ideas from webly-supervised learning, where the focus is on harnessing the vast and noisy data available on the web have lent ideas for large pre-training Guo et al. (2018; 2017); Dai et al. (2018). In Beser & Bulling (2021) and Li et al. (2018) webly supervised zero-shot learning in natural language processing is explored. Further, Zhang et al. (2017) demonstrates how web data can enhance fine-grained categorization tasks. Zang et al. (2019); Zhang et al. (2017) combines webly supervised and zero-shot learning techniques for fine-grained recognition tasks. Together, these works offer valuable insights into leveraging web data for training machine learning models. However, using search engines where text is the medium for representation remains under-explored.

## 3 BACKGROUND: MULTIMODAL PRE-TRAINED MODELS

A large pre-trained model such as CLIP is trained using a self-supervised objective to align the output (*i.e.*, representations) obtained from both language and image outputs, respectively. The significance of this is that when trained (unsupervised), we can map either textual or visual inputs using their corresponding neural encoders $\phi_t, \phi_v$ to the same semantic space (assuming the outputs are normalized). One way to interpret these models is to consider an image $\boldsymbol{x} \in \mathcal{X}$ and its corresponding textual description $\boldsymbol{t} \in \mathcal{T}$ as $p(\boldsymbol{x}, \boldsymbol{t}) \propto \exp(\langle \phi_x(\boldsymbol{x}), \phi_t(\boldsymbol{t}) \rangle)$. Here, $\mathcal{X}, \mathcal{T}$ denote the space of image and text collected in a large corpus. Then, for the downstream classification task with the label set $\mathbf{Y}$, we have:

$$ p_{\text{pre}}(\boldsymbol{y} \mid \boldsymbol{x}) = \frac{\exp\left(\langle \phi_x(\boldsymbol{x}), \phi_t(\boldsymbol{y}) \rangle\right)}{\sum_{\boldsymbol{y}' \in \mathbf{Y}} \exp(\langle \phi_x(\boldsymbol{x}), \phi_t(\boldsymbol{y}') \rangle)}, \quad \boldsymbol{y}^\star = \arg\max_{\boldsymbol{y}' \in \mathbf{Y}} \quad p_{\text{pre}}\left(\boldsymbol{y}' \mid \boldsymbol{x}\right), \tag{1} $$

where the labels $\boldsymbol{y}$ are used in their textual form, benefiting from the inherent capability of such pre-trained models for open-ended problems.

## 4 PROACTIVE LEARNING

Our main goal is to improve the performance of a pre-trained model at inference time, *i.e.* CLIP. In our setting, we are given an (unlabelled) "target" dataset $\mathbf{X}$ of images and a list of potential

labels $\mathbf{Y}$. We are interested in training a complementary model on a "small" retrieved dataset to predict the target labels. To build such a dataset, we leverage the vast amount of data on the internet by creating a retrieval mechanism (note there is no curated labelled training set). We propose a retrieval mechanism that considers the uncertainty in the classification of the pre-trained model, queries and downloads images from search engines like Google. Our retrieval mechanism aims to effectively cover the uncertain classes in the target dataset. The retrieved dataset is expected to be noisy, containing unrelated images for certain class names. For example, one can look after images of flowers with the class name 'Prince of Wales feathers' and retrieve images of royalty instead of flowers. In light of this, our retrieval mechanism also incorporates a refinement process to clean the dataset before training a classifier. We consider the following steps as the general algorithm:

1. determine the uncertain instances in the given unlabelled target dataset (see Sec. 4.1)
2. formulate a query and invoke the search engine to retrieve the relevant images (see Sec. 4.2)
3. refine and filter out the unrelated images (see Sec. 4.3)
4. train a small model (*e.g.* a linear probe) for classification (see Sec. 4.4)

Formally, for the set of uncertain labels $\mathbf{Y}_{\text{uncertain}}$, *at inference time*, with access to a search engine $q_{\text{se}}$ that allows sampling $n$ most appropriate images given a query $\boldsymbol{y}_r$, we consider

$$\boldsymbol{\theta}^\star = \arg\max_{\boldsymbol{\theta}} \frac{1}{n} \sum_{\boldsymbol{x}_r, \boldsymbol{y}_r} \log \left( p_{\boldsymbol{\theta}}(\boldsymbol{y}_r \mid \boldsymbol{x}_r)^{\mathbb{I}[f(\boldsymbol{x}_r \mid \mathbf{X})]} \right), \quad \boldsymbol{x}_r \sim q_{\text{se}}(\boldsymbol{x}_r \mid \boldsymbol{y}_r), \boldsymbol{y}_r \sim p(\boldsymbol{y}_r \mid \mathbf{Y}_{\text{uncertain}}), \quad (2)$$

where $\boldsymbol{\theta}$ is a small number of parameters compared to the ones in the pre-trained model (*e.g.* a simple classifier optimized on a frozen CLIP backbone) and $f(\boldsymbol{x}_r \mid \mathbf{X})$ is a function indicating whether a given sample $\boldsymbol{x}_r$ belongs to the same distribution as those in $\mathbf{X}$ (see Sec. 4.3). The optimal inference-time $\boldsymbol{\theta}^\star$ are the maximum likelihood parameters for the set $\{(\boldsymbol{x}_r, \boldsymbol{y}_r)\}$ obtained from the search engine. Effectively, a search engine enables us to *sample from the distribution* of all images potentially relevant to a query $\boldsymbol{y}_r$.

## 4.1 UNCERTAINTY IN PRE-TRAINED MODELS

To construct the dataset, we focus on the image samples that the pre-trained model is uncertain about its label. Considering the conditional probability in Eq. (1), we can use the Shannon entropy as a measure of uncertainty, with a threshold $\tau_H$ to construct a subset of uncertain instances *i.e.*

$$\mathbf{X}_{\text{uncertain}} = \left\{ \boldsymbol{x} \mid \mathbb{H}_{\text{pre}}(\boldsymbol{y} \mid \boldsymbol{x}) \geq \tau_H, \forall \boldsymbol{x} \in \mathbf{X} \right\}. \quad (3)$$

Note that this uncertainty measures the uniformity of predictions for the given labels using the pre-trained model. If the model is not confident, *i.e.* produces a higher score for a class, this uncertainty will be high. We collect samples whose entropy is higher than a threshold needing additional information for prediction. We pick all the class labels in the top-$k$ predictions for the uncertain instances $\boldsymbol{x} \in \mathbf{X}_{\text{uncertain}}$ and create an uncertain label set $\mathbf{Y}_{\text{uncertain}}$.

## 4.2 CREATING A SEARCH ENGINE-BASED DATASET

Now that we have the set of uncertain samples, we can use those instances for constructing queries aiming to retrieve useful images from the Internet. We propose the following to generate a query:

1. **Classnames** ($p^{\text{cls}}$): Directly using the "$\{$class_name$\}$".

2. **Descriptions** ($p^{\text{desc}}$): Using an LLM, *i.e.* GPT-3, to generate descriptions for the class names similar to the approach in Menon & Vondrick (2023). We concatenate class names with these descriptions as "$\{$class_name$\}$ which (is/has/etc) $\{$descriptor$\}\}$".

3. **Captioning** ($p^{\text{cap}}$): Using a captioning method such as BLIP (Li et al., 2022b). The class name is then concatenated with this caption as "$\{$class_name$\}$ $\{$caption$\}$". Note that in this approach, the query is constructed for each instance individually as opposed to the previous two alternatives.

We can now create a dataset for each query option. For instance, for **Classnames**, we have:

$$\mathbf{D}_{\text{uncertain}}^{\text{cls}} = \left\{ (\boldsymbol{x}_r, \boldsymbol{y}_r) \,\Big|\, \boldsymbol{x}_r \in \underbrace{\mathbf{SearchEngine}(\boldsymbol{t}, n), \boldsymbol{t} \sim p^{\text{cls}}(\boldsymbol{t} \mid \boldsymbol{y}_r)}_{\sim q_{\text{se}}(\boldsymbol{x}_r \mid \boldsymbol{y}_r)} \quad \forall \boldsymbol{y}_r \in \mathbf{Y}_{\text{uncertain}} \right\}, \quad (4)$$

where $n$ indicates the number of top images to be queried. Similarly, we construct $\mathbf{D}_{\text{uncertain}}^{\text{desc}}, \mathbf{D}_{\text{uncertain}}^{\text{cap}}$. We use $\mathbf{D}_{\text{uncertain}}$ to denote any of these datasets.

## 4.3 REFINEMENT

Ideally, the samples in this set should consist of images from a distribution similar to that of the target dataset, yet possessing distinctive features that can enhance performance. However, either due to query ambiguity or dataset specificity, there might be images in the retrieved set $\mathbf{D}_{\text{uncertain}}$ that do not conform to the distribution of those in $\mathbf{X}$.

Therefore, a refinement step is imperative to effectively dispose of the potentially noisy samples to create a refined subset $\mathbf{D}_{\text{refined}}$. This necessitates a comparison between the distributions of the retrieved and target datasets. Instead of adopting complex density estimation models, such as those necessitating the training of additional neural networks (*e.g.*, (Kingma & Welling, 2013; Abbasnejad et al., 2020; 2019; Ho et al., 2020)), we note CLIP's image embeddings, like many other contrastive learning alternatives, are inherently normalised to the unit hypersphere (Shi et al., 2023) (see Fig. 2). Thus, we leverage the mixture of von Mises-Fisher Distributions (MoVMF) Banerjee et al. (2005) to model this hypersphere distribution. The mixture of von Mises-Fisher distributions is a statistical model that combines multiple von Mises-Fisher distributions to represent data distributed on a hypersphere in high-dimensional space, allowing for the modelling of complex directional data patterns. Once optimised, MoVMF allows us to represent the target distribution as $p(\boldsymbol{x} \mid \mathbf{X}) = \sum_i \pi_i p_{\boldsymbol{\mu}_i, \boldsymbol{\kappa}_i}(\boldsymbol{x})$, where $\pi_i$ represents the mixing coefficient, and $\boldsymbol{\mu}_i$ and $\boldsymbol{\kappa}_i$ are the parameters of the von Mises-Fisher

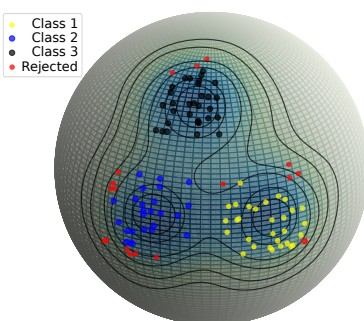

Figure 2: Conceptual visualisation of images on the unit hypersphere. Images from three classes are represented by dots, with the colourmap indicating probability density from a mixture of von Mises-Fisher distributions. Threshold lines demonstrate selection boundaries, with images beyond these lines (in red) deemed too distant for selection.

Distribution. We then utilise the most likely component to assign a hard cluster for each instance. Subsequently, we only keep samples that are sufficiently close to the mean of each von Mises-Fisher component, *i.e.*

$$f(\boldsymbol{x}_r \mid \mathbf{X}) = \mathbb{I}\big[\langle \phi_x(\boldsymbol{x}_r), \boldsymbol{\mu}_{i^*}\rangle < \tau_R\big], \quad i^* = \arg\max_i p_{\boldsymbol{\mu}_i, \boldsymbol{\kappa}_i}(\boldsymbol{x}_r). \tag{5}$$

We explore two alternative approaches (see appendix), but empirical results demonstrate MoVMF outperforms them. It is straightforward to implement and imposes minimal assumptions on the data.

## 4.4 PREDICTION

In the final step, we train a classifier $p_{\boldsymbol{\theta}}$ on the refined retrieved samples. However, if there is no information gained (*i.e.* information gain or mutual information of label and additional data is not positive), we fallback to the pre-trained model. In other words, we only use this new model if it leads to a reduced entropy (*i.e.* the confidence in predictions improves):

$$\boldsymbol{y}^* = \arg\max_{\boldsymbol{y}} \begin{cases} p_{\boldsymbol{\theta}^\star}(\boldsymbol{y} \mid \boldsymbol{x}), & \text{if } \text{IG}(\boldsymbol{y} \mid \boldsymbol{x}, \mathbf{D}_{\text{refined}}) \leq 0 \\ p_{\text{pre}}(\boldsymbol{y} \mid \boldsymbol{x}), & \text{otherwise} \end{cases}, \tag{6}$$

where $\boldsymbol{y}^*$ is the predicted for $\boldsymbol{x}$, $\text{IG}(\boldsymbol{y} \mid \boldsymbol{x}, \mathbf{D}_{\text{refined}}) = \mathbb{H}_{\text{pre}}(\boldsymbol{y} \mid \boldsymbol{x}) - \mathbb{H}_{\boldsymbol{\theta}^\star}(\boldsymbol{y} \mid \boldsymbol{x}, \mathbf{D}_{\text{refined}})$ is the information gain and $\mathbb{H}_{\boldsymbol{\theta}^\star}(\boldsymbol{y} \mid \boldsymbol{x}, \mathbf{D}_{\text{refined}})$ denotes the entropy of the classifier $p_{\boldsymbol{\theta}^\star}$ (*i.e.* the entropy once the model is updated).

## 5 EXPERIMENTS

We employ CLIP (Radford et al., 2019) as the pre-trained model to validate our approach. Our focus primarily lies in a zero-shot scenario, wherein we work under the assumption of not having access to a training dataset. Consequently, our initial comparison revolves around assessing the performance against the baseline zero-shot CLIP. Nevertheless, we also provide supplementary experiments exploring alternative zero or few-shot settings, which can be found in the appendix for a more comprehensive understanding of our approach. It's crucial to note that our methodology refrains from utilizing any labelled training data except in explicitly specified cases.

|  | Flowers | Pets | Cars | Food | ImageNet |
|---|---|---|---|---|---|
| *ZeroShot CLIP-B-16* | 71.15 | 89.04 | 64.71 | 88.73 | 68.33 |
| Ours | 86.32 ↑15.17 | 92.78 ↑3.74 | 81.82 ↑17.11 | 88.97 ↑0.24 | 70.14 ↑1.81 |
| LP w/ Training Dataset | 96.41 | 92.72 | 85.64 | 92.08 | 78.66 |
| LP w/ $D_{uncertain}$ + Training dataset | 96.57 ↑0.16 | 93.62 ↑0.9 | 86.03 ↑0.39 | 92.15 ↑0.07 | 78.94 ↑0.28 |

Table 1: Our results of training a **LinearProbe** (LP) on the retrieved dataset $D_{uncertain}$, the train split of the target dataset and the combination of these two.

## 5.1 DATASETS AND METRICS

We evaluate our method on five popular image classification datasets: Flowers Nilsback & Zisserman (2008), Pets Parkhi et al. (2012), Stanford Cars Krause et al. (2013), Food Bossard et al. (2014) and ImageNet Deng et al. (2009). These datasets consist of 102, 37, 196, 101 and 1000 classes. Our objective is to retrieve a dataset that matches the distribution of the target dataset, acknowledging the need for CLIP for more information. We evaluate the impact of the retrieved dataset $D_{uncertain}$ by training a linear probe and combining its predictions with those made by CLIP using the accuracy metric. Moreover, we visually compare the distribution of the target and retrieved dataset using UMAP. We evaluated the overlap between the retrieved and the target dataset, Appendix A, our datasets $D_{uncertain}^{cls}$, $D_{uncertain}^{cap}$ and $D_{uncertain}^{desc}$ exhibit a 0% of overlap with the target testing dataset.

## 5.2 RESULTS AND ANALYSIS

In Table 1, we show (Top) the accuracy results of ZeroShot (ZS) CLIP and our proposed method with Linear Probing (*LP*) trained on the best-retrieved dataset. Also, we show (Bottom) the accuracy of these methods when they are trained on the training set of each target dataset and its combination with our retrieved dataset. We can see our method consistently outperforms ZS CLIP. We observe a significant improvement in Flowers and Cars datasets with over 15% in accuracy, showing that the samples retrieved from the search engine best helps with the features needed to perform well on the test set of these datasets. When we use our method on Food and ImageNet, we obtain a slight improvement of 0.24% and 1.81%, respectively. These two are challenging datasets in terms of constructing queries. In the case of Food, we speculate due to diversity and variability of the food images online and specificity of this dataset, the improvements remained marginal.

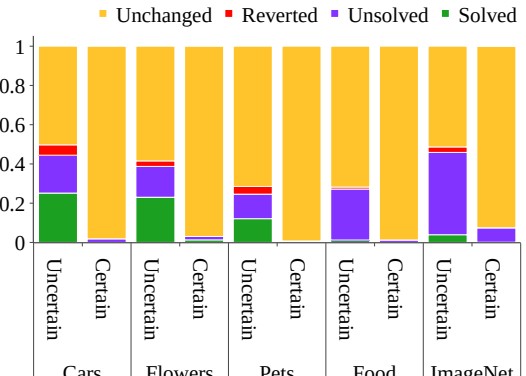

Figure 3: The impact of our method on uncertain and certain subsets: **Solved** shows the percentage of cases incorrectly predicted before and corrected. **Unsolved** cases are incorrectly predicted before and after. **Reverted** cases refer to cases correctly predicted before but wrongly predicted after we applied our method. **Unchanged** cases were correctly predicted before fine-tuning and are still correctly predicted afterwards.

Regarding ImageNet, a significant challenge arises from the fact that certain classes necessitate the inclusion of highly specialized types of images. Take, for instance, classes representing specific dog species. When attempting to collect images from the web for these classes, we encounter two prevalent issues: a considerable increase in variability within the images or a prevalence of highly specific, limited-content images. Also, we found possible misinterpretations of the search engines in terms of concepts. Some category names could confuse search engines. For example, for the category name *Agama* in ImageNet, which for Google could be an Indian religion or the lizard that we are looking for. In the case of Pets, we obtain a significant improvement of more than 3% of accuracy, surpassing the performance of training a Linear Probe on the training set of the target dataset.

Moreover, when we use the concatenation of our retrieved dataset with the training set of the target dataset, we observe the retrieved dataset consistently improves the performance of the baseline methods when they only have access to the training set. It indicates the retrieved images provide information that supplements that of the training distribution. This suggests our approach can easily be integrated into the current pipelines.

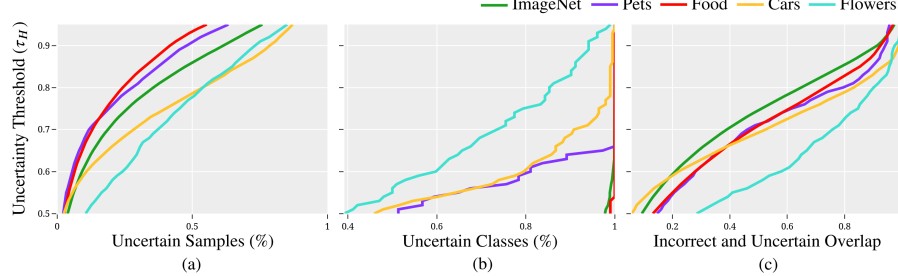

Figure 4: Uncertain plots for each dataset: (a) percentage of uncertain instances, (b) the ratio of uncertain classes, and (c) the overlap between incorrect and uncertain instances, versus on the uncertainty threshold $\tau_H$.

| $\mathbf{D}^{\text{cls}}_{\text{uncertain}}$ | $\mathbf{D}^{\text{cap}}_{\text{uncertain}}$ | $\mathbf{D}^{\text{desc}}_{\text{uncertain}}$ | Refined | Flowers | Pets | Cars | Food | ImageNet |
|---|---|---|---|---|---|---|---|---|
| — | *ZeroShot CLIP-B-16* | | — | 71.15 | 89.04 | 64.71 | 88.73 | 68.33 |
| ✔ | | | | 86.24 ↑15.09 | 92.61 ↑3.57 | 80.52 ↑15.81 | 88.97 ↑0.24 | 69.05 ↑0.72 |
| ✔ | | | ✔ | 86.32 ↑15.17 | 92.64 ↑3.60 | 80.48 ↑15.77 | 88.35 ↓-0.38 | 65.61 ↓-2.72 |
| ✔ | ✔ | | | 85.44 ↑14.29 | 92.45 ↑3.41 | 80.33 ↑15.62 | 88.89 ↑0.16 | 69.54 ↑1.21 |
| ✔ | ✔ | | ✔ | 86.27 ↑15.12 | 92.56 ↑3.52 | 81.82 ↑17.11 | 88.87 ↑0.14 | 69.78 ↑1.45 |
| ✔ | | ✔ | | — | 92.59 ↑3.55 | — | 88.87 ↑0.14 | 69.80 ↑1.47 |
| ✔ | ✔ | ✔ | | — | 92.78 ↑3.74 | — | 88.86 ↑0.13 | 70.14 ↑1.81 |

Table 2: Accuracy results by training our method with the different retrieval techniques $\mathbf{D}^{\text{cls}}_{\text{uncertain}}$, $\mathbf{D}^{\text{cap}}_{\text{uncertain}}$, $\mathbf{D}^{\text{desc}}_{\text{uncertain}}$ and the effect of our refinement procedure.

Figure 3 shows a more detailed comparison of CLIP predictions before and after update using retrieved images. In this bar plot, we can see that the slight improvement in the Food dataset is due to the small margin in the reverted cases relative to the solved cases. This is while reverted cases make up a much smaller percentage of the changes in the uncertain cases of other datasets. Moreover, the effect of our method is minimal in certain cases where the percentage of unchanged cases makes up the largest fraction of every dataset. In other words, adding a Linear Probe to deal with uncertain instances does not affect the instances where CLIP is certain about the prediction.

## 5.3 ON THE DIFFERENT RETRIEVED DATASETS

Table 2 shows the accuracy results of the proposed query constructions $\mathbf{D}^{\text{cls}}_{\text{uncertain}}$, $\mathbf{D}^{\text{cap}}_{\text{uncertain}}$ and $\mathbf{D}^{\text{desc}}_{\text{uncertain}}$, Section 4.2, and the effect of the refinement process. In the case of $\mathbf{D}^{\text{desc}}_{\text{uncertain}}$, we use the descriptions shared by Menon & Vondrick (2023). However, they only shared descriptions for the Flowers, Cars and ImageNet datasets. The effect of the refinement process does not always improve the performance of the initial retrieved dataset. Excluding Food, it always worked for datasets constructed with more diverse queries like $\mathbf{D}^{\text{cap}}_{\text{uncertain}}$. The images tend to be more diverse and cover more of the real distribution of each label when we retrieve the $\mathbf{D}^{\text{cap}}_{\text{uncertain}}$ and $\mathbf{D}^{\text{desc}}_{\text{uncertain}}$. However, those procedures also add much more noisy instances. Using the refinement, we can get more samples and simultaneously remove ambiguous images that could deteriorate the performance of our method.

Except for the retrieved datasets $\mathbf{D}^{\text{cls}}_{\text{uncertain}}$ with refinement for Food and ImageNet, we obtain improvements compared to the performance of zero-shot CLIP-B-16 when we use the retrieved dataset. This is noticeable for the cases of Flowers and Cars, where our method obtains a large accuracy improvement on average of 14.92% and 16.01%, respectively.

## 5.4 COMPARISON WITH INTERNET EXPLORER

We compare our method with Internet Explorer (IE) Li et al. (2023). In contrast with our method, which retrieved data only for the uncertain classes of CLIP and trained only with the retrieved dataset, Internet Explorer uses MoCo-V3 He et al. (2019) with ResNet50 backbone pre-trained on ImageNet and also the training split of the target dataset. In Table 3,

|  | Flowers | Pets | Food | × Images |
|---|---|---|---|---|
| *ZS CLIP-ResNet-50* | 66.22 | 85.72 | 80.97 | |
| Internet Explorer | **99.10** | 90.80 | 84.60 | ×10⁶ |
| Ours | 90.39 | **91.17** | **85.97** | ×10⁴ |

Table 3: Comparison with Internet Explorer: Our approach excels in the Food and Pets dataset with far fewer retrieved images. Here, × Images indicates the order of magnitude difference in retrieved images.

we compare the accuracy results of Flowers, Pets and Food since they do not report on ImageNet-1K and Cars. To fairly compare the two methods, we use CLIP with ResNet-50 backbone and train the LP with the combination of the training set of the target dataset and our retrieved dataset $\mathbf{D}_{\text{uncertain}}^{\text{cls}}$. Using two orders of magnitude of fewer data, we outperform IE on the Pets and Food dataset by 0.37% and 1.37%. In the case of Flower, our zeroshot baseline has a much lower performance (66.22%) than their baseline (94.6%). Thus, we obtain a larger margin improvement (24%) than IE (4.5%) on the Flower benchmark.

## 5.5    On the value of the threshold $\tau_H$

To study the effects of the Shannon confidence threshold $\tau_H$ value on the uncertainty quantification process described in Section 4.1, we perform experiments on the training set of the target dataset. Figure 4 (c) shows a correlation between the incidence of incorrectly predicted samples considered uncertain for a particular $\tau_H$ and the total number of incorrectly predicted samples. This correlation highlights the utility of employing uncertainty as a surrogate measure for identifying instances of incorrect predictions. Remarkably, this relationship holds consistently across all the datasets. To maximize the capture of incorrectly predicted samples using uncertainty as a proxy, selecting a threshold value near 0.9 would effectively achieve this objective. Moreover, Figure 4 (b) shows the percentage of uncertain classes as a factor of the threshold $\tau_H$. While the proportion of uncertain samples and their overlap in the Food and ImageNet datasets closely resembles that of Pets, Flowers, and Cars (as shown in Figure 4 (c)), it is noteworthy that the proportion of uncertain classes in the former two datasets nears 100% (as depicted in Figure 4 (b)). This observation implies that, for Food101 and ImageNet, there is at least one uncertain instance for each class across a wide range of confidence thresholds. In contrast, the remaining

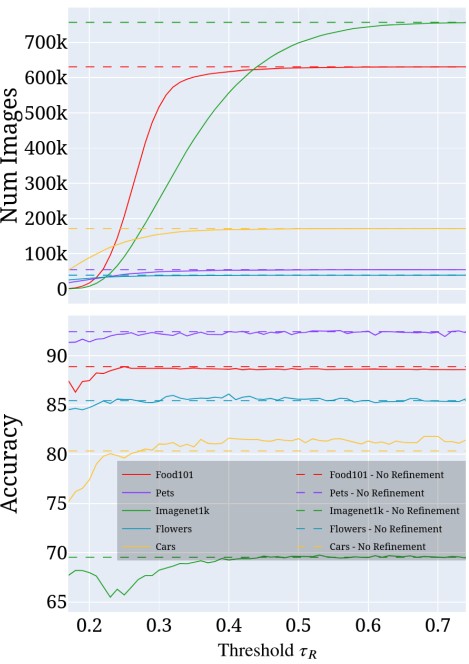

Figure 5: Effect of the threshold $\tau_R$ in the number of images and accuracy

datasets exhibit a more gradual increase in uncertainty as the confidence threshold $\tau_H$ changes, indicating that the model is more confident of some specific classes.

## 5.6    On the refinement process

**The value of threshold $\tau_R$.** Figure 5 shows the effect of a strict to a more relaxed refinement threshold $\tau_R$ on the accuracy of our refinement process in the retrieved dataset $\mathbf{D}_{\text{uncertain}}^{\text{cap}}$. Notice that small values in $\tau_R$ mean we reduce the acceptance range, Section 4.3. We decide to use a threshold of 0.45 across all the datasets, where we get a good improvement in performance in a variety of benchmarks whilst not removing a big portion of the retrieved dataset, thus covering the distribution of the target dataset, Appendix B.

**Qualitative analysis of refinement** Figure 6 shows random samples of our refinement process's accepted and rejected instances for Pets and Food datasets. Other datasets are in the Appendix J. We can see that the refinement process can reject images that contain words related to the class names on the dataset but do not necessarily have the object in interest, thus not providing information to our classifier. It is also capable of removing instances that contain the object, like Siamese cats, but they are out of the distribution of the training dataset.

## 5.7    On using Parameter-Efficient Fine-Tuning

LoRA Hu et al. (2022) is a prominent parameter-efficient fine-Tuning method primarily employed in transformers to avoid the need for retraining the entire model for downstream tasks. Instead of

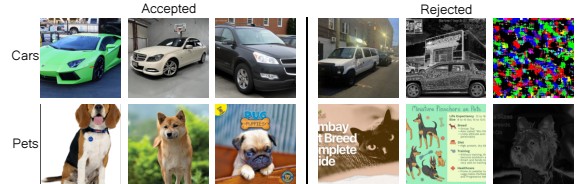

Figure 6: Random samples of the accepted and rejected instances of the refinement process

|  | Flowers | Pets | Cars | Food |
|---|---|---|---|---|
| *ZS CLIP-B-16* | 71.15 | 89.04 | 64.71 | 88.73 |
| *Using LAION-5B* | 61.86 ↓-9.29 | 80.84 ↓-8.20 | 56.60 ↓-8.10 | 88.35 ↓-0.38 |
| *LP w/ $\mathbf{D}^{cls}_{uncertain}$* | 86.24 ↑15.09 | 92.61 ↑3.57 | 80.52 ↑15.81 | 88.97 ↑0.24 |
| *LoRa w/ $\mathbf{D}^{cls}_{uncertain}$* | 84.29 ↑13.14 | 92.56 ↑3.52 | 80.55 ↑15.84 | 88.98 ↑0.25 |

Table 4: Accuracy results of training a LP vs using **LoRa** on $\mathbf{D}^{cls}_{uncertain}$ and also the results of using LAION-5B to retrieved images and train our LP.

retraining the entire model from scratch for each task, LoRA preserves the pre-trained model and introduces smaller, trainable matrices to each layer of the model. These matrices enable the model to adjust to various tasks without modifications to all of the parameters. We use LoRa to adapt CLIP with our retrieved dataset $\mathbf{D}^{cls}_{uncertain}$ as an alternative to LP. Table 4 shows the performance of CLIP-LoRa. As observed, LP outperforms CLIP-LoRa in both Flowers and Pets benchmarks. In the cases of Cars and Food, the performance gain is marginal. Remarkably, using the retrieved dataset $\mathbf{D}^{cls}_{uncertain}$ always helps LoRa to adapt. This indicates more parameters of LORA compared to LP overfits to the retrieved dataset.

## 5.8 ON AN ALTERNATIVE TO A SEARCH ENGINE USING LAION-5B

Instead of using the language to create queries and search images, we propose an alternative approach that uses the LAION-5B Schuhmann et al. (2022) to retrieve similar images in the multi-modal space. Since we cannot access the training set to get the labels of the most uncertain cases, we retrieved images from the most certain image for CLIP. In this case, we aim to retrieve the neighbour images similar to those of certain instances so that we can assign the label of the certain sample to the retrieved one. As shown in Table 4, naively retrieving similar images and training a linear probe on those images does not help improve predictions on the target dataset.

## 6 CONCLUSION

The proposed approach of leveraging search engines for machine learning is a novel idea with connections to different areas of research, including active learning, weakly supervised learning, cross-modal retrieval, and language models. In the future, we envision a dynamic approach where pre-trained models can access the internet to acquire new knowledge as they encounter novel concepts for prediction. This stands in contrast to traditional task-specific static models, which are expected to perform adequately even in the presence of previously unseen instances, a challenge that current out-of-distribution generalization research strives to address.

Much like the art of effective prompt engineering, our methodology suggests that formulating well-constructed search queries (attainable *e.g.* through interaction with LLMs) may yield improved performance, warranting further exploration. It is worth noting that our approach may encounter limitations in scenarios with restricted online resources, such as specialized medical domains, where purpose-built search engines could be advantageous.

One avenue for potential future research involves amalgamating results from various search engines. This could lead to divergent sets of retrieved data. Additionally, the susceptibility to bias is a pertinent concern when dealing with internet-sourced datasets, as they often lack rigorous refinement. As these methods progress toward maturity, addressing such issues becomes increasingly important.

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

## A EVALUATING POSSIBLE DATA LEAKS

A major apprehension when using search engines to build a dataset revolves around the potential performance enhancement being predominantly driven by retrieving a significant portion of the test set images. To tackle this issue, we mitigate these concerns by assessing the extent to which the retrieved images match those found in the test set of the target dataset by employing a hash comparison method. We first convert each image in the target and retrieved sets to its byte representation and hash it using the SHA-256 algorithm, resulting in a unique identifier for the individual content of each image. The number of overlapping hash codes, serving as a quantitative measure of data leakage between the datasets, are reported in Table 5. Our method retrieves a minuscule portion of test set images online, thus no leakage. Our method's substantial consistent improvement cannot be attributed to test set leakage, implying that its enhanced performance stems from improved feature learning and generalization.

|  | Flowers | Pets | Cars | Food | ImageNet |
|---|---|---|---|---|---|
| Target test set size | 6148 | 3669 | 8041 | 25250 | 50000 |
| $\mathbf{D}^{cls}_{uncertain}$ | 0 (0.0%) | 0 (0.0%) | 0 (0.0%) | 0 (0.0%) | 1 ($\approx$ 0.0%) |
| $\mathbf{D}^{cap}_{uncertain}$ | 0 (0.0%) | 1 ($\approx$ 0.0%) | 0 (0.0%) | 0 (0.0%) | 0 (0.0%) |
| $\mathbf{D}^{desc}_{uncertain}$ | ——— | 3 ($\approx$ 0.0%) | ——— | 0 (0.0%) | 0 (0.0%) |

Table 5: The count of leaked images from the test set is determined using image hashing. This calculation involves assessing the proportion of test images found within the collection of images downloaded. We provide details for each dataset, including the test set's size, the number of leaked test images, and the corresponding percentage of the test set represented in blue.

## B UMAP VISUALIZATION OF THE TARGET DATASET AND THE RETRIEVED DATASET

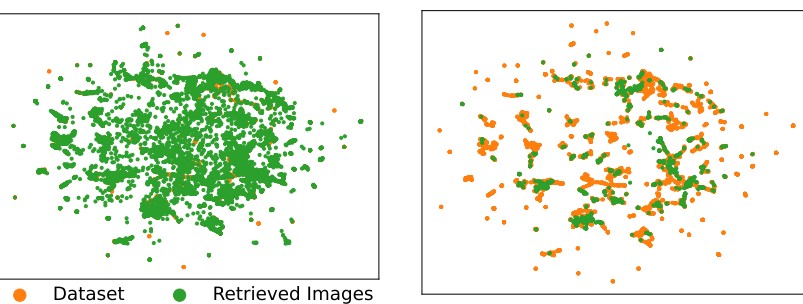

● Dataset    ● Retrieved Images

Figure 7: UMAP visualization of the distribution of the image features from CLIP-B-16 for the training split of the original and retrieved dataset. It shows the effect of the refinement method on the retrieved dataset from a less (left) to a more strict (right) refinement.

Figure 7 shows the UMAP visualization of the image features of the target dataset and the retrieved dataset for two choices of the threshold $\tau_R$. First, we can see that the retrieved dataset matches the distribution of the training set. The retrieved dataset can fill the gap in the distribution of each class. This explains why our LinearProbe can improve the performance of CLIP in uncertain cases. Moreover, we can see how a more strict threshold $\tau_R$ retained less number of instances in the retrieved set. A sweet spot could be found when the retrieved dataset covers the biggest portion of the target distribution and removes the noisy samples.

## C EFFECTS OF PROMPT SPECIFICITY ON PREDICTION CONFIDENCE

Since predictions from CLIP are not only reliant on the input images but also the text prompt, the uncertainty of the prediction will also be affected by this text. To study the effects of the specificity

of the input prompt on the confidence in the prediction, we use six methods of prompting, each at different levels of descriptiveness. The results of this study are visualized in Fig. 8.

The prompt named CLS refers to using class names with the format "{class_name}" to prompt CLIP. ZS expands this approach by adding a generic description of the dataset in the format "{class_name}, a type of {description}", e.g. the prompt used for Pets is "{class_name}, a type of pet". For GPT3, a description of the class name is generated by OpenAI GPT3 first, and the description is used for prompting without the addition of the class name that was used to generate the description or any other phrases. Wordnet and Wikidef refer to the definition of the class name in WordNet Miller (1998) and Wiktionary Wiktionary Contributors (2023). Similar to GPT3, no additional phrases were added for prompting. Finally, Hi refers to generating a prompt using the WordNet hierarchy where the last three words in the ontological tree leading to the class name are concatenated and used as the prompt.

Figure 8 indicates that utilizing ZS individually, or in combination with CLS, marginally enhances the network's confidence in predictions. This enhancement is attributable to a more refined representation of the prompts, thereby reducing text ambiguity. A notable observation is the significant reduction in network confidence with the use of GPT-3 descriptions compared to simpler prompts such as CLS or ZS. An examination of these descriptions revealed that GPT-3 prompts tend to be more narrative and detailed rather than focusing on keyword specificity. That is, instead of representing a concise combination of the class's crucial characteristics, these prompts offer extensive narratives, often overlooking the visually distinguishable aspects of the class.

## D  ON THE NUMBER OF IMAGES PER QUERY

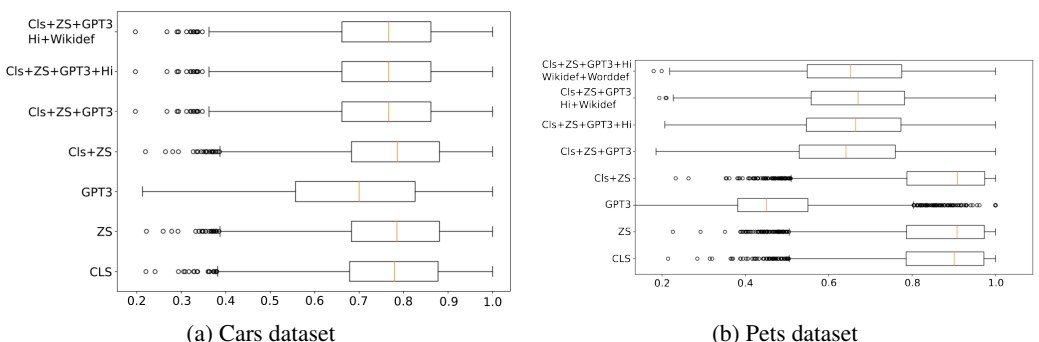

(a) Cars dataset                              (b) Pets dataset

Figure 8: Boxplot of confidence values when prompts with varying specificity are used. **CLS** refers to using class names from the dataset for prompting. **ZS** are prompts specific to each dataset described in more detail in Appendix C. **GPT3**, **Wikidef**, **Hi** and **Worddef** are prompt templates taken from Li et al. (2022a). **GPT3** refers to descriptions of each class generated by OpenAI-GPT3 Zhao et al. (2023) and used as the prompt. **Worddef** and **Wikidef** are Wordnet Miller (1998) and Wiktionary Wiktionary Contributors (2023) definitions of each class in the dataset (With the exception of Cars for which no classes have Wordnet definitions). **Hi** refers to a set of words extracted from Wordnet based on the ontological hierarchy of the class name.

To investigate the effect of the number of images retrieved per class, we train a linear probe on subsets of increasing size from $\mathbf{D}^{cls}_{\text{uncertain}}$ and report the accuracy results for each dataset in Fig. 9. For Cars, Flowers and Pets we observe a steady increase in accuracy as the number of used images from the retrieved set increases. This is while the lowest number of used images (25 for all datasets) still outperforms the zero-shot accuracy for every dataset except Food. ImageNet on the other hand, sees a slight dip in accuracy when using 50 images, but the overall observed trend is upward with more images.

Additionally, for the Cars, Flowers and Pets datasets, the change in accuracy relative to the change in number of images slows down with more images and sees a plateau as we reach 100 images. This could potentially be because of the lack of information in the newer images or because the search engine cannot provide more useful images for the classifier.

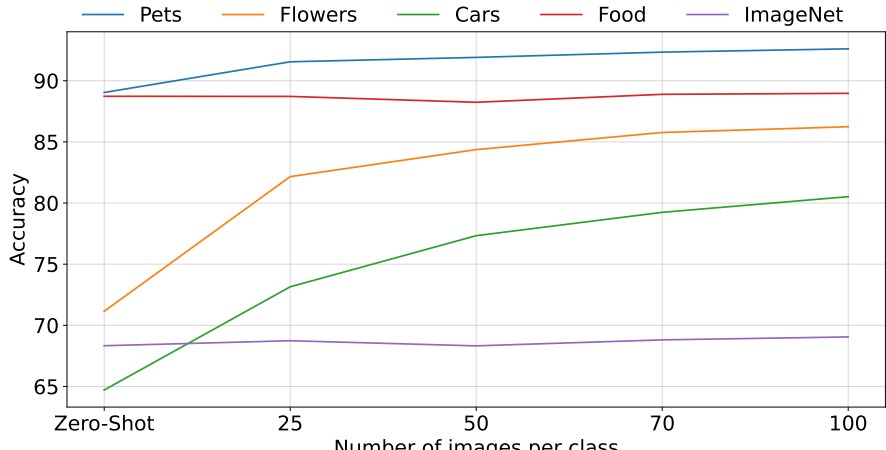

Figure 9: Number of images

# E    OTHER REFINEMENT METHODS

The refinement method described in Section 4.3 shows the exclusion of unrelated and noisy images from the retrieved set; however, this method is intrinsically dependent on the training split of the target dataset. In this experiment, we investigate an alternative refinement approach that is not dependent on accessing images from the dataset and explore an approach that circumvents the computationally demanding clustering strategy depicted in Section 4.3.

For refining retrieved images, we need anchor points that facilitate gauging the similarity of each retrieved image to the dataset distribution. Even without access to images from the target dataset, text from the labels of the target dataset can still serve as a viable alternative. Essentially, this boils down to performing inference on the retrieved images, finding the text with the most similarity to each retrieved image and rejecting the image if this value is lower than a threshold $\tau_R$ similar to the subsequent equation.

$$f(\boldsymbol{x}_r \mid \boldsymbol{y}) = \mathbb{I}\big[\langle \phi_x(\boldsymbol{x}_r), \phi_t(\boldsymbol{y}_{i^*}) \rangle < \tau_R\big], \quad i^* = \arg\max_i \langle \phi_x(\boldsymbol{x}_r), \phi_t(\boldsymbol{y}_{i^*}) \rangle. \tag{7}$$

On the other hand, as a substitute for the clustering on the unit hypersphere methodology detailed in Section 4.3, we investigate the application of the more efficient k-means clustering. Recognizing that k-means is not inherently designed for clustering on a sphere, we start by elevating each point on the hypersphere to the tangent space at the mean of the image modality. This mean is obtained using the images from the training split of the target dataset. Consequently, the overall procedure for this strategy is similar to the one described in Section 4.3 with the exception of employing the Euclidean tangent space, thereby facilitating clustering using k-means. The outcome of the text-based approach is visualized in Fig. 11 and the results of the k-means based approach are visualized in Fig. 10.

While the approach proposed in Section 4.3 gives better results than the ones seen in Fig. 11 and 10, we still see some improvement especially for Cars in both figures. Two observed issues with the tangent space clustering are the issues with the distortion in the projection of images to the tangent space and the other is the problems with failure of k-means in finding appropriate cluster centers leading to large dips in the accuracy observed in Fig. 10a.

# F    IMAGE DISTRIBUTIONS COMPARISON

To qualitatively investigate the differences between the distribution of images from the target dataset to that of the retrieved dataset, we train a classifier to detect which dataset the images are coming from. In particular, the labels for training this classifier are 0 if the image is coming from the target dataset and one if the image is from the retrieved set. The outputs of this classifier are shown in

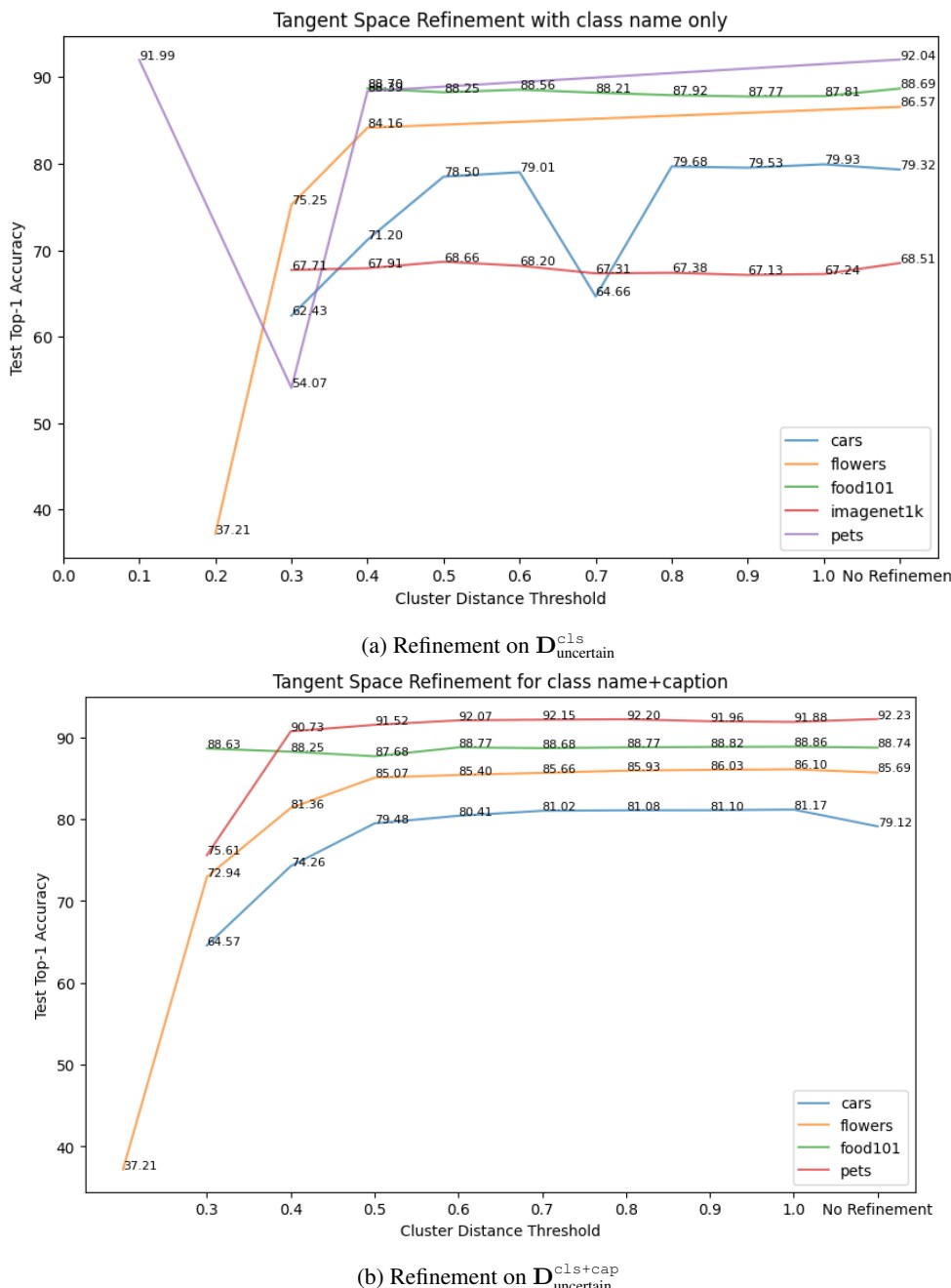

Figure 10: Refinement using the tangent space of the image modality with k-means clustering

Fig. 12. It is evident that there is some information, especially in the case of Food and Pets datasets, that allows the classifier to easily distinguish the retrieved images from the target images. Our hypothesis is that this easy distinction by the classifier is because of a combination of reasons, one of which is a genuine shift between the distribution of retrieved and target images. This is because of the lack of specificity in the class names used to retrieve images. For example, when retrieving images of cars that a certain manufacturer makes, the images of the same cars with the same make and model from the retrieved and target dataset will not look different since they are made by the same manufacturer and the class names are descriptive enough for the images to match the ones in the target dataset. On the other hand, when it comes to the Food or Pets dataset, images retrieved by a search engine will be much more diverse than the Food and Pets datasets, which are curated

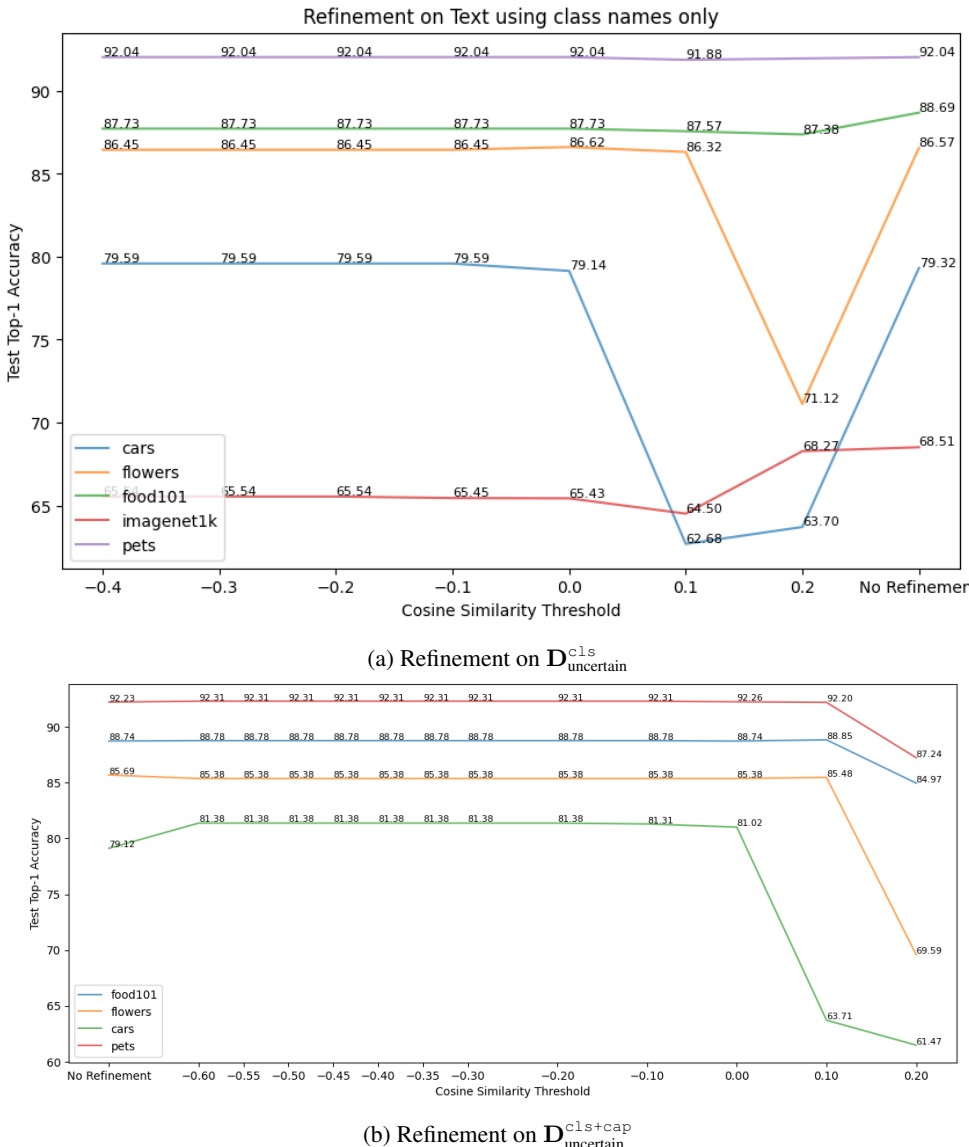

(a) Refinement on $\mathbf{D}_{\text{uncertain}}^{\text{cls}}$

(b) Refinement on $\mathbf{D}_{\text{uncertain}}^{\text{cls+cap}}$

Figure 11: Refinement using the labels of the target dataset

datasets with biased distributions. Therefore, distribution shifts are expected for these datasets. The Flowers dataset seems to be in the middle of the other three datasets since both a good overlap and many distinct images exist between the retrieved and target datasets.

Another potential reason for this distinction could be much shallower than the previously suggested one. The images from the target dataset are curated images that commonly have a fixed height-to-width ratio. On the other hand, retrieved images will have different ratios even between the retrieved images themselves. During the resizing and cropping performed before passing the images through CLIP, it is possible that some content is always cropped out from the retrieved images while all the content of the object in the image from the target dataset is always present. This could be a shallow identifier for the classifier to use to perform shortcut learning.

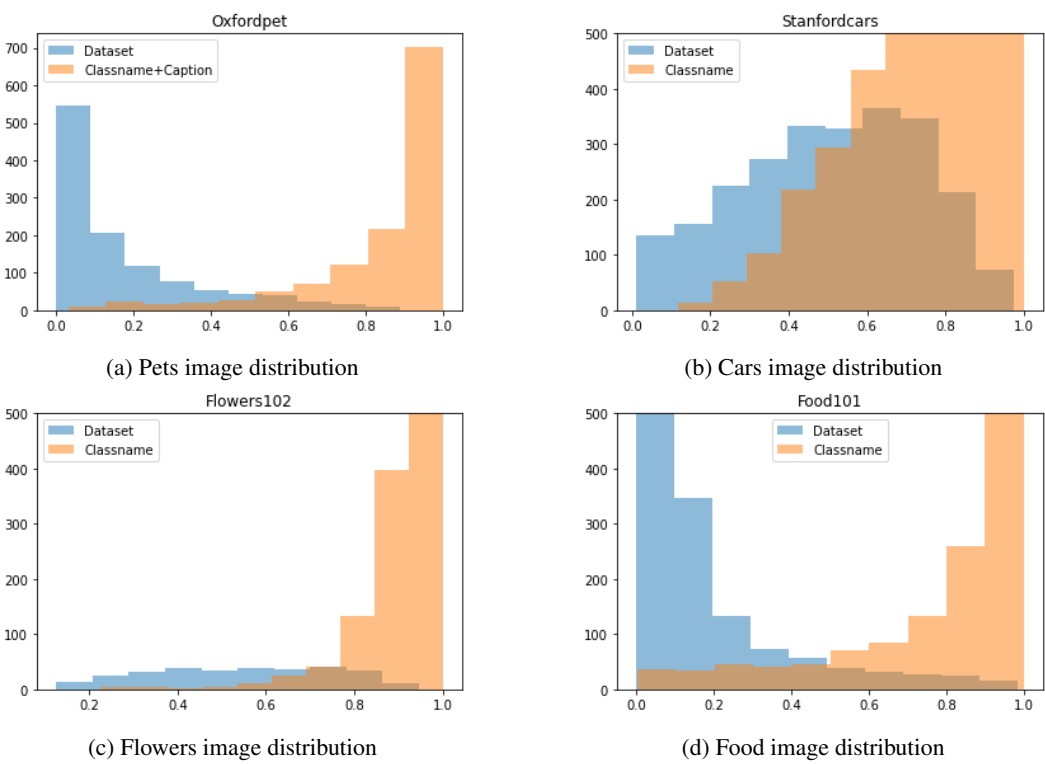

(a) Pets image distribution

(b) Cars image distribution

(c) Flowers image distribution

(d) Food image distribution

Figure 12: Distribution of the retrieved images compared to the images from the target dataset.

|  | Flowers | Pets | Cars | Food | ImageNet |
|---|---|---|---|---|---|
| *Zero Shot CLIP ViT-g-14 LAION-2B* | 77.35 | 94.33 | 92.92 | 91.55 | 76.64 |
| Ours | 86.53 ↑9.18 | 94.41 ↑0.81 | 93.73 ↑0.81 | 91.61 ↑0.06 | 77.12 ↑0.48 |
| *Zero Shot CLIP ViT-L-14 OpenAI* | 79.17 | 93.43 | 77.94 | 97.36 | 75.53 |
| Ours | 87.88 ↑8.71 | 94.28 ↑0.85 | 85.76 ↑7.82 | 97.24 ↓0.12 | 76.06 ↑0.53 |

Table 6: Performance of different CLIP backbones by using our method. We can see a consistent performance improvement in each of the datasets.

## G   DIFFERENT BACKBONES OF CLIP

We try our retrieved datasets $\mathbf{D}^{\texttt{cls}}_{\texttt{uncertain}}$ with different backbones of CLIP,*i.e.* ViT-g-14 and ViT-L-14. Obtaining a consistent improvement with respect to the zero-shot version across the datasets. Except for Food, which in the case of ViT-L-14 is already reaching 97% of accuracy. We believe the main reason that we do not gain an improvement, in this case, is that CLIP ViT-L-14 already covers most of the distribution of Food, and our dataset cannot provide extra valuable information. Table 6

## H   OTHER METHODS WITH OUR DATASET

### H.1   TIP-ADAPTER

Tip-Adapter Zhang et al. (2021) is an adaptation technique designed for CLIP, enabling few-shot classification without extensive training. It inherits the training-free advantage seen in zero-shot CLIP while also delivering competitive performance compared to methods that demand extensive training. Tip-Adapter constructs an adapter using a key-value cache model based on the few-shot training set. This adapter then updates the prior knowledge encoded in CLIP through feature retrieval. Moreover, by fine-tuning the cache model, Tip-Adapter can achieve state-of-the-art performance with significantly fewer training epochs than existing methods, demonstrating both effectiveness and efficiency. We chose this method to test our dataset since it is a training-free technique that can show how relevant it is to have a dataset that matches the distribution of the target dataset. We performed experiments using Tip-Adapter with the retrieved dataset $\mathbf{D}^{\texttt{cls}}_{\texttt{uncertain}}$. These experiments are performed 100 times per dataset with different seeds. Thus, using different images for each k-shot. We use $k = 2, 4, 8.16$ instances from the retrieved dataset. Figure 13 shows the performance in each benchmark. We can see that Tip-Adapter, in its two versions (Training-Free and Fine-Tunning), benefits from our retrieved dataset, showing that our methodology to automatically construct a dataset that covers the distribution of the target dataset works not only in our Linear Probe.

## I   QUANTITATIVE MEASURE OF DIFFERENCE BETWEEN RETRIEVED AND TARGET DATASETS

To quantify the difference between the distribution of features of the retrieved images compared to that of the target dataset, we use the Earth Mover's Distance (EMD) Rubner et al. (2000). EMD or the Wasserstein distance between the two distributions can show how different the retrieved images are in the feature space compared to the target dataset through quantifying the minimum cost of turning one distribution into the other, where cost is measured in terms of moving mass through the space. This value for Pets, Cars, Flowers and Food datasets is visualized in Fig. 14. Comparing the datasets to each other, Flowers has the lowest difference between the retrieved and target dataset while Pets has the highest difference. When crossing these values with the results in Table 1, no correlations are observed between the EMD and obtained increase in accuracy given that the EMD for the Food dataset lies between datasets that get large improvements. However, when comparing these results with Fig. 7, it is evident that the retrieved images are mostly close to the clusters formed by the images from the target dataset. This means that while the Wasserstein distance between the retrieved and target datasets will be low, the bump in obtained accuracy potentially has to do with retrieving images that lie on the tails of the image distributions especially for datasets such as Food characterized by a diverse array of images.

## J    QUALITATIVE ANALYSIS OF REFINEMENT

To give a more intuitive sense of what images are rejected or accepted during the refinement process we provide the visualizations in Fig. 15, Fig. 16 and Fig. 17. These visualizations show random samples from the retrieved dataset that were accepted/rejected by the refinement procedure. A common occurrence in the retrieval process is the retrieval of images of only text that is related to the queried class name but does not provide any information in the content of the image that is similar to the class name that was originally searched for. Since we use the CLIP features for the refinement procedure, by thresholding the similarity between the features of the retrieved and original dataset, the unrelated images can be effectively rejected.

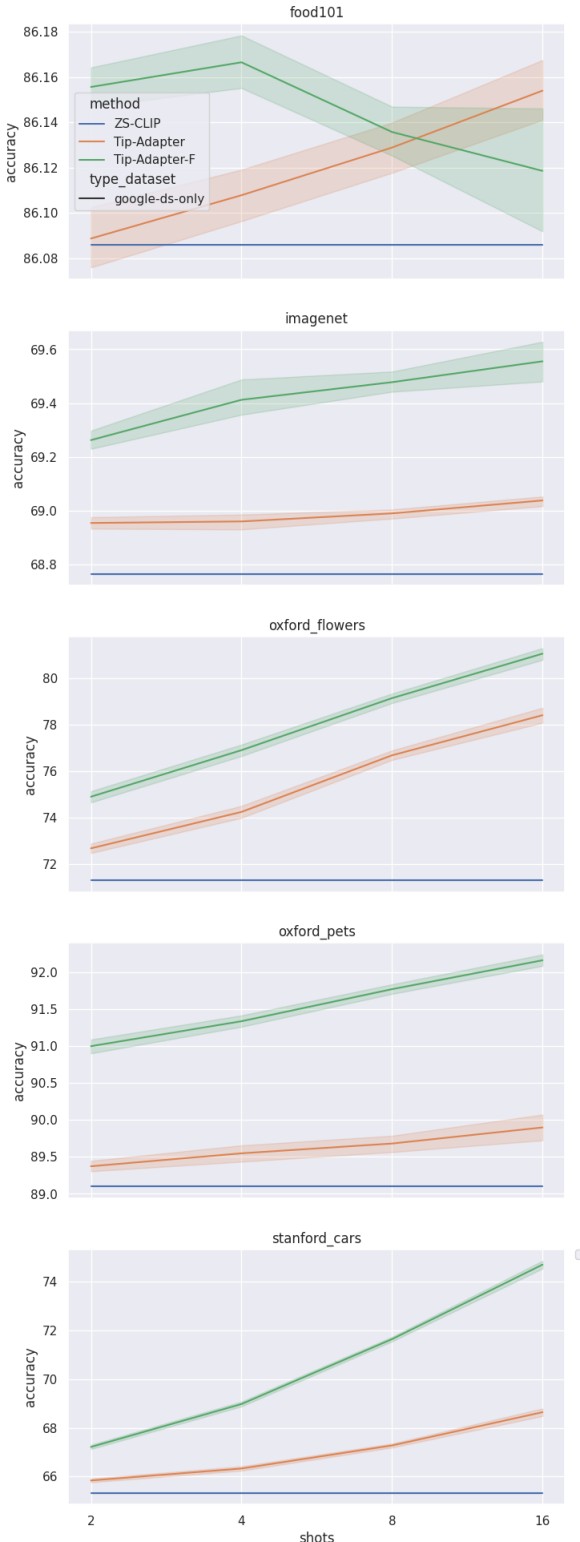

Figure 13: Few-shot experiments using Tip-Adapter Zhang et al. (2021) with our retrieved dataset $\mathbf{D}^{\texttt{cls}}_{\text{uncertain}}$

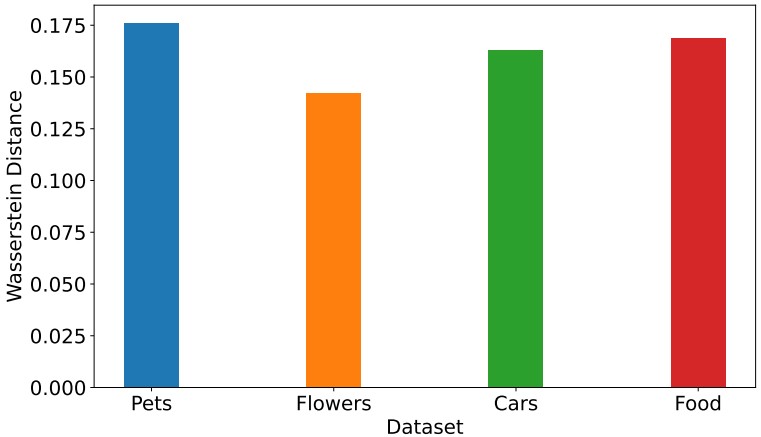

Figure 14: EMD between the feature distribution of the retrieved and target images

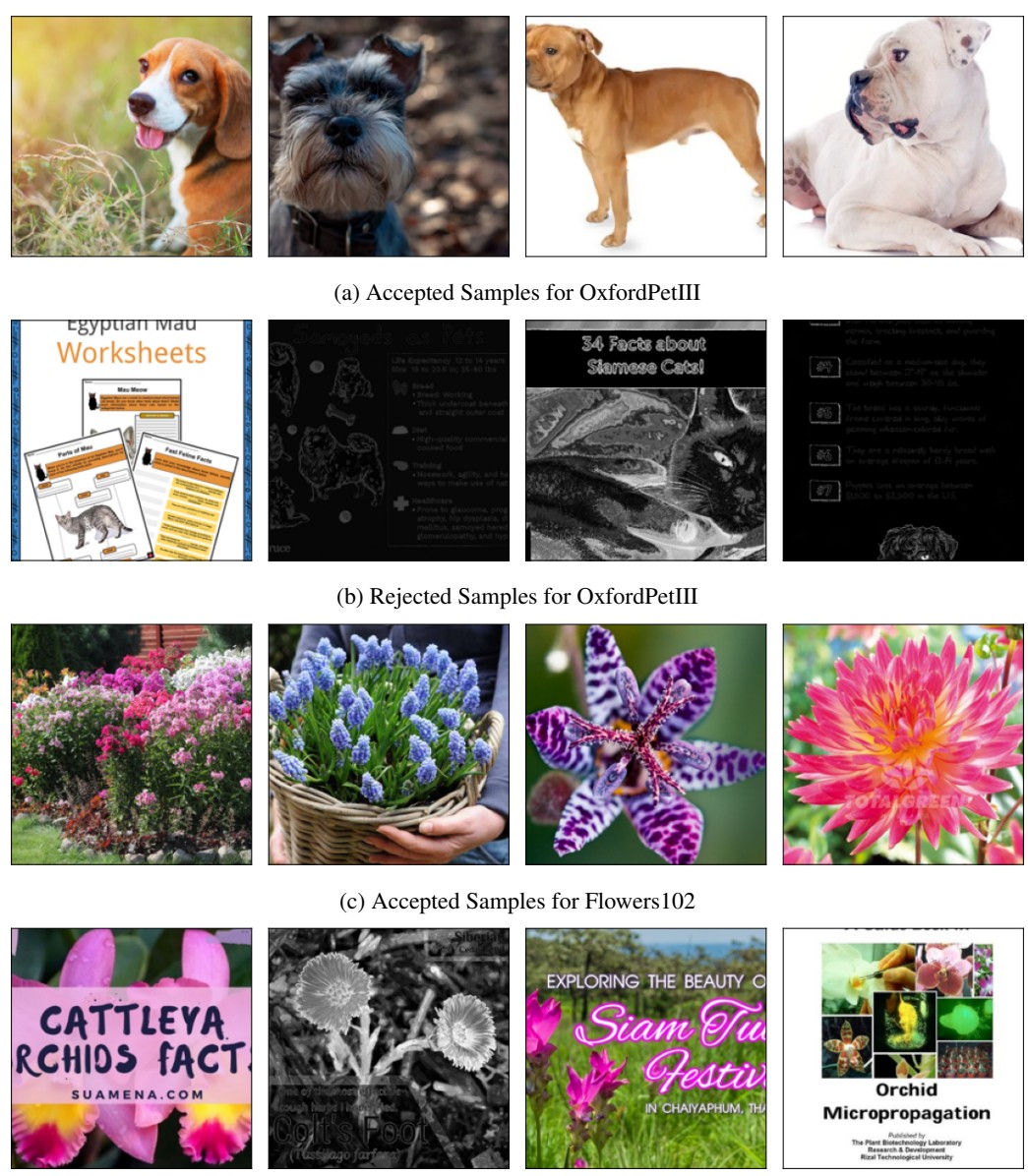

(a) Accepted Samples for OxfordPetIII

(b) Rejected Samples for OxfordPetIII

(c) Accepted Samples for Flowers102

(d) Rejected Samples for Flowers102

Figure 15: Random samples from each retrieved dataset based on the refinement procedure described in Section 4.3

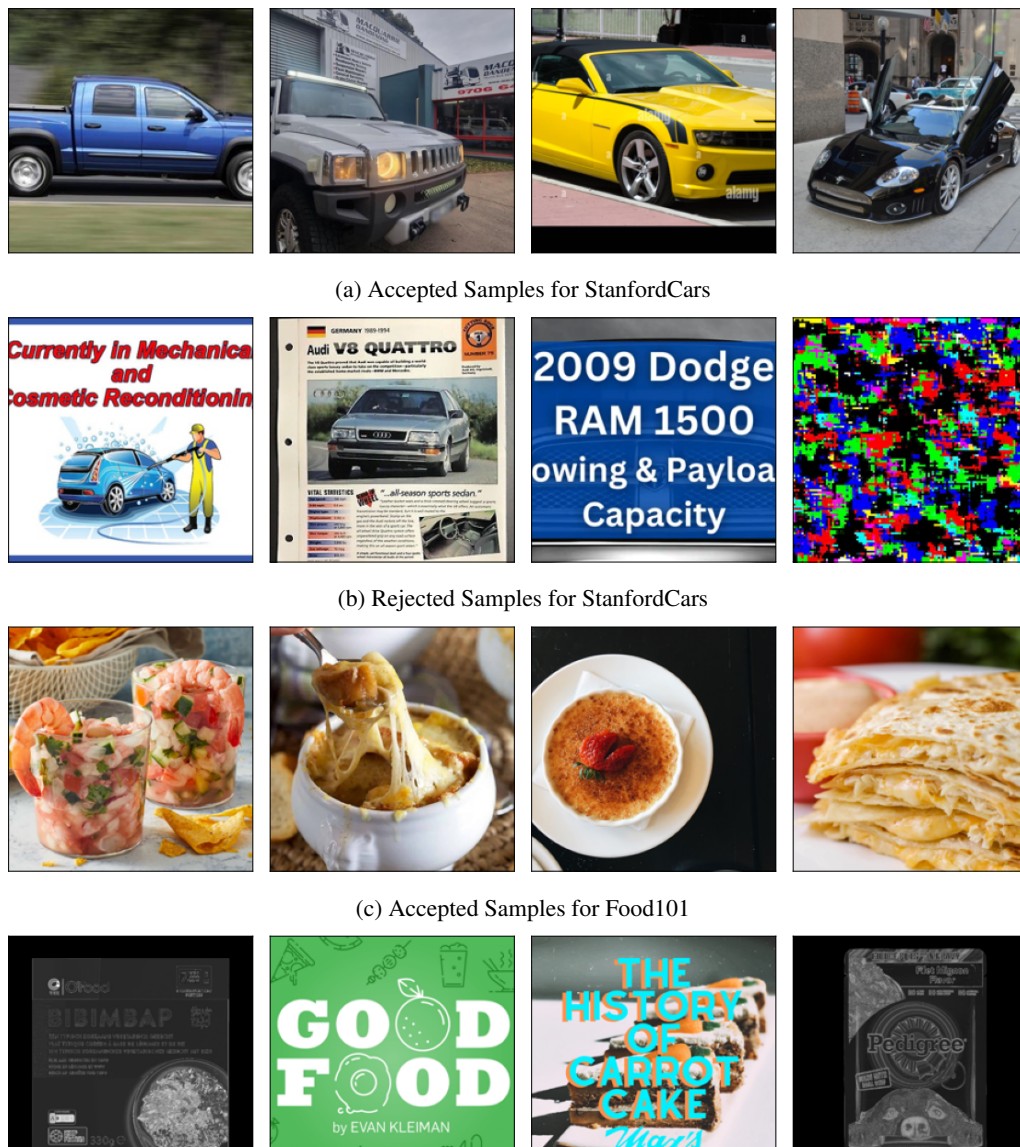

(a) Accepted Samples for StanfordCars

(b) Rejected Samples for StanfordCars

(c) Accepted Samples for Food101

(d) Rejected Samples for Food101

Figure 16: Random samples from each retrieved dataset based on the refinement procedure described in Section 4.3

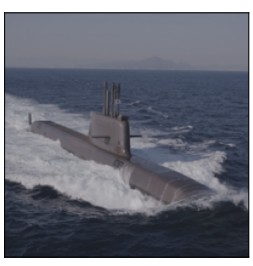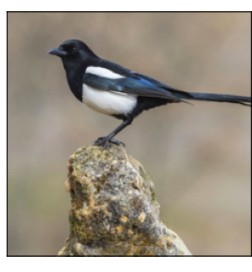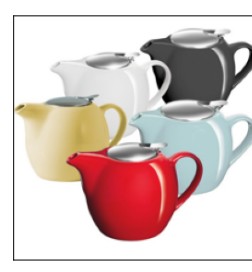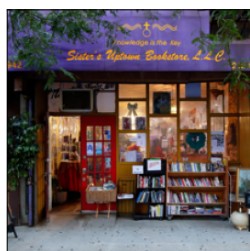

(a) Accepted Samples for ImageNet

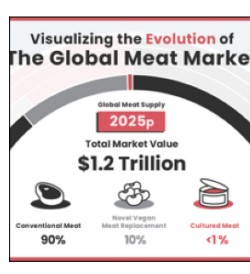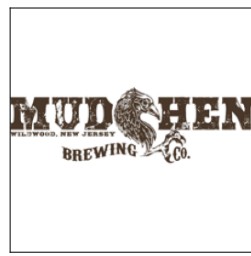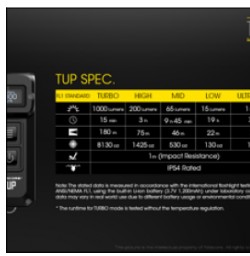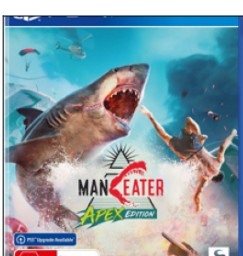

(b) Rejected Samples for ImageNet

Figure 17: Random samples from each retrieved dataset based on the refinement procedure described in Section 4.3

