# OpenReview forum: "Proactive Learning: Search-augmented learning using Pre-trained Models"
_ICLR.cc/2024/Conference — ICLR 2024 Conference Withdrawn Submission_

### Official Review · Reviewer_dJXS · 2023-10-27

**Soundness:** 3 good
**Presentation:** 3 good
**Contribution:** 2 fair
**Rating:** 3
**Confidence:** 4

**Summary:**

The paper proposes a method to use web-search to retrieve images in order to enrich a pre-trained model for visual recognition. The approach encompasses 4 steps: 1) Determine “uncertain” samples in the target dataset. 2) Use the top-K class labels for the uncertain samples in order to formulate queries and retrieve images from a search engine. 3) Cluster the retrieved images and discard images far from cluster means. 4) Train a small model (e.g., linear probe) for classification.

Experiments are provided to show the proposed method is able to successfully retrieve images that lead to accuracy gains (sometimes significant) w.r.t. the accuracy of the pre-trained model.

**Strengths:**

**S1.** The method is simple and easy to reproduce.

**S2.** The method addresses an important problem / use case.

**S3.** The experimental validation suggests the method is (potentially) practically useful.

**Weaknesses:**

**W1.** Important design choices are not properly discussed, supported, or validated:
- In my view, the key element in the method are the queries used to retrieve (useful) images. However, the three query generation approaches are not specified in detail. Moreover, the only experimental comparison of the three approaches shows minor performance differences between all of them (Table 2). This is confusing.
- Details regarding hyperparameter selection and effects are lacking, e.g., number of queries, number of retrieved images, number of mixture components, etc.

**W2.** If the model is overconfident and wrong about a class the proposed method (specifically the uncertainty estimation) will not retrieve images and, thus, will not enrich the model.

**W3.** It is stated the retrieved dataset used in Table 1 (which provides the main results) has been somehow determined to be the “best” (section 5.2, first sentence). However, it was not stated how this determination was made.

**W4.** Also in Table 1, the improvement over training with the Training Dataset (last two rows) is most often marginal.

**W5.** The comparison with Internet Explorer in Table 3 is difficult to interpret since the methods are using different pre-trained models (MoCo-V3 vs CLIP).

**Questions:**

**Q1.** How could the method recover in case retrieved images actually lead to performance degradation?

**Q2.** How was it determined the dataset used in Table 1 was the “best”?

**Q3.** How could the method be utilized without access to any labeled data?

**Q4.** In table 3 the proposed method hurt performance significantly (once). Why and how can this be avoided?

**Details Of Ethics Concerns:**

None.

---

### Official Review · Reviewer_6yvj · 2023-10-29

**Soundness:** 3 good
**Presentation:** 3 good
**Contribution:** 3 good
**Rating:** 6
**Confidence:** 4

**Summary:**

This paper proposes a pipeline for augmenting the pre-trained model at test time through web search. Specifically, it includes four steps: determine uncertain instances, web search, data filtering and training. Experiments are performed with the CLIP model on five visual recognition datasets.

**Strengths:**

1. The overall web search-based framework in this work is clear, reasonable and novel.
2. The problem this work tries to solve is meaningful.
3. Refining coarse and noisy data by leveraging a mixture of von Mises-Fisher distributions is interesting.
4. The improvements over Flowers and Cars datasets are significant.
5. Experiments under different settings are thorough.

**Weaknesses:**

1. The refinement process does not work well on Food and ImageNet datasets. Is there any explanation for this?

2. In Figure 4, when using $D_{uncertain}^{cls}$, $D_{uncertain}^{cap}$, $D_{uncertain}^{desc}$ together, why exclude the results with refinement?

3. Experiments with LoRA are interesting. However, only $D_{uncertain}^{cls}$ case is included. experiments on ImageNet are missing.

4. The caption of Table 1 is not appropriate. Since the method uses the combined prediction (as stated in Section 4.4), it may mislead readers if calling the method "LinearProbe (LP)".

**Questions:**

See the Weaknesses part for the major questions. I have some other questions as follows.
1. Does the web search-based method can be extended to a continual learning setting?
2. Would the method fail if using a non-contrastive pre-trained model?

---

### Official Review · Reviewer_QgSd · 2023-11-08

**Soundness:** 2 fair
**Presentation:** 3 good
**Contribution:** 2 fair
**Rating:** 3
**Confidence:** 4

**Summary:**

The method in this paper uses on-line retrieval from the web to augment a training set with additional data in a transfer learning scenario.  Starting by training an initial classifier on a target dataset (either with labels, or using zero-shot if no labels), low-confidence classes/images are identified using entropy thresholds.  Additional data for these classes are retrieved using a search engine queried using the class name, and additional generated text (e.g. generated class descriptions or image captions).  Irrelevant images are identified and discarded based on proximity to the target data distribution, modeled by a VMF mixture on image embeddings.  The method yields significant gains in the unlabeled setting, as small gains even when using all target dataset labels.

**Strengths:**

This is a simple retrieval mechanism, and its value is demonstrated well in the setting proposed.  In addition to overall gains, the profile of where there are increases/decreases (Fig 3) is a good description.  Evaluations are performed on several appropriate datasets, showing consistent behavior.

**Weaknesses:**

While the benefit of including images from relevant searches is clear, many of the mechanisms introduced here aren't yet demonstrated to be effective.  See below for additional details, but in particular:

* Uncertainty threshold $\tau_H$ is chosen to be quite large, perhaps including most classes.
* Searching for class names alone appears to have similar gains compared to the other text queries, except in one or two cases
* The refinement step shows little gain in most cases

Taking the above points together, it appears simply taking top n images retrieved for each class query may perform about as well, but I didn't see this baseline in the comparisons.

**Questions:**

Additional details relating to my above comments:

Sec 5.4  I like this comparison, but it's difficult to compare apples-to-apples.  IE uses ImageNet and the target dataset for unsupervised pretraining (it doesn't use the target datasets' labels at any point afaict).  Does the "Ours" line in Table 3 train on the target datasets' labels?  The text indicates it does ("train the LP with the combination of the training set of the target dataset and our retrieved dataset").  If so, this should be broken down to also use linear probe on CLIP without retrieved data (just using the target dataset), and vice-versa, using the retrieved data only (without using the target dataset labels).  The most relevant comparison to IE would be the one that doesn't use the target dataset labels; it may underperform it in this case --- though it's a much simpler system, which has other advantages.


Sec 5.5.  $\tau_H=0.9$ looks like it would include most classes.  What is the accuracy vs different values for this threshold?  Does limiting the number of classes here have any gain in accuracy, compared to querying all classes?  If not, how much does it limit additional computation?


Fig 5:  The dataset to see most accuracy increase from refinement is Cars (yellow), which goes from around 80 to 82 here.  But this is true even at $\tau_R = 0.7$ at the right-hand side of the plot, where the corresponding num images (yellow) coincides with the dashed line not using refinement.  How can this be the case?  How many samples are discarded for these larger refinement thresholds (and what about thresholds larger than 0.7, going up to 1.0)?


Sec 4.4:  How often does each case (using refined model vs original model) occur at test time?  What about if the target set labels are included?


Smaller comments/questions:


ImageNet --- this was originally collected using search queries containing the class names (though, over 10 years ago).  Does it really have 0 overlap?  I see appendix A uses sha256 for comparison --- that's a good check, but I'm not sure it's sufficient:  The images may have been re-encoded when the dataset was compiled, so won't match byte-for-byte against their original source.

Eq 5:  Why use dot product with a threshold rather than thresholding on the VMF prob for the max prob component, or other certainty measure using the probs that also account for $\kappa$ in their fit?

Is it possible to produce a large class imbalance for the final classifier training, if many more samples for some classes are retained compared to others?  Similarly, do any of the non-uncertain classes become worse or less certain after including retrieved images?



Eq 2 suggests y_r are drawn from a distribution --- how are these sampled or weighted in the sum, if p(y_r) is anything other than uniform over a finite set?

Eq 2:  I[f] in the exponent is confusing --- this can be multiplied outside the log.

Sec 4.1:  "i.e. produces a higher score" --- If the score is 1 while other classes' scores are 0, then entropy will also be low, so this is a little bit confusing here

Eq 4:  Drawing from the distribution $t ~ p^{cls}$, suggests more than one $t$ is used for each $y_r$, but it isn't clear how many.  If there is only one for each of p^cls and p^desc, then I think drawing t from p is overly complex to describe this set.  If multiple description samples are drawn, it's not clear how many.  This formula also doesn't include the different (original) $x$ instances in the uncertain classes, needed to generate captions.  Still, I understand what is going on.